# The Use of Kidney Biomarkers, Nephrin and KIM-1, for the Detection of Early Glomerular and Tubular Damage in Patients with Acromegaly: A Case–Control Pilot Study

**DOI:** 10.3390/diseases12090211

**Published:** 2024-09-11

**Authors:** Iulia Stefania Plotuna, Melania Balas, Ioana Golu, Daniela Amzar, Roxana Popescu, Ligia Petrica, Adrian Vlad, Daniel Luches, Daliborca Cristina Vlad, Mihaela Vlad

**Affiliations:** 12nd Department of Internal Medicine, Discipline of Endocrinology, “Victor Babes” University of Medicine and Pharmacy, 300041 Timisoara, Romania; iulia.plotuna@umft.ro (I.S.P.); balas.melania@umft.ro (M.B.); golu.ioana@umft.ro (I.G.); amzar.daniela@umft.ro (D.A.); vlad.mihaela@umft.ro (M.V.); 2Department of Endocrinology, County Emergency Hospital, 300723 Timisoara, Romania; 3Center for Molecular Research in Nephrology and Vascular Disease, “Victor Babes” University of Medicine and Pharmacy, 300041 Timisoara, Romania; popescu.roxana@umft.ro (R.P.); petrica.ligia@umft.ro (L.P.); 4Department of Microscopic Morphology, Discipline of Cellular and Molecular Biology, “Victor Babes” University of Medicine and Pharmacy, 300041 Timisoara, Romania; 52nd Department of Internal Medicine, Discipline of Nephrology, “Victor Babes” University of Medicine and Pharmacy, 300041 Timisoara, Romania; 6Department of Nephrology, County Emergency Hospital, 300723 Timisoara, Romania; 72nd Department of Internal Medicine, Discipline of Diabetes, Nutrition and Metabolic Diseases, “Victor Babes” University of Medicine and Pharmacy, 300041 Timisoara, Romania; 8Department of Diabetes, Nutrition and Metabolic Diseases, County Emergency Hospital, 300723 Timisoara, Romania; 9Department of Sociology, Western University of Timisoara, 300223 Timisoara, Romania; daniel.luches@e-uvt.ro; 10Biochemistry and Pharmacology Department, Discipline of Pharmacology, “Victor Babes” University of Medicine and Pharmacy, 300041 Timisoara, Romania; vlad.daliborca@umft.ro; 11Clinical Laboratory, County Emergency Hospital, 300723 Timisoara, Romania

**Keywords:** acromegaly, renal biomarkers, KIM-1, nephrin, diabetes mellitus

## Abstract

Background: Acromegaly is a rare disorder caused by excessive growth hormone (GH) secreted from a pituitary tumor. High levels of GH and insulin growth factor-1 can lead to renal hypertrophy, as well as to diabetes mellitus and hypertension, which negatively impact kidney function. It is believed that high GH may also be involved in the onset of diabetic nephropathy, the main cause of end-stage kidney disease in developed countries. Material and methods: This case–control study was conducted on 23 acromegalic patients and on a control group represented by 21 healthy subjects. The following parameters were determined for all the subjects: serum creatinine, serum urea, estimated glomerular filtration rate (eGFR), urinary albumin/creatinine ratio (UACR), nephrin and kidney injury molecule 1 (KIM-1). Results: Patients with acromegaly showed higher levels of UACR and lower levels of eGFR as compared to healthy subjects. No significant correlations were found between clinical or biochemical parameters associated with acromegaly and nephrin or KIM-1. Conclusions: There was no glomerular or proximal tubular damage at the time of the study, as proven by the normal levels of the biomarkers nephrin and KIM-1. Studies including more patients with uncontrolled disease are needed to clarify the utility of nephrin and KIM-1 for the detection of early kidney involvement in acromegalic patients.

## 1. Introduction

Acromegaly is a rare endocrinological disorder nearly always caused by a growth hormone (GH)-secreting pituitary adenoma [1]. Pituitary adenomas develop from a single cell, accounting for approximately 10% of primary intracranial tumors [1]. This hormonal excess leads to increased insulin growth factor-1 (IGF-1) and the abnormal growth of tissues and organs, including the kidneys [2]. The GH/IGF-1 axis has numerous sites of action in the renal system, and its effects are mediated mainly through its intracellular receptor STAT5 [2]. Due to the important physiological role of the GH/IGF-1 axis in the development of tissues and organs, animal models were used to study the mechanisms involved in renal development [3]. Mice lacking the GH receptor had smaller kidneys than controls, proving that GH contributes to renal ontogenesis [3]. This aspect was also noted in uni-nephrectomized experimental mice, where GH and locally secreted IGF-1 induced compensatory renal hypertrophy [4]. The GH receptor is expressed in the podocytes, mesangial cells and tubules, where it maintains normal sodium and water balance (probably through its action on the epithelial sodium channel) and increases the glomerular filtration rate (eGFR) [3]. An increased eGFR seems to play a role in the development of albuminuria [5]. Some data pointed out that GH might induce the expression of transforming growth factor-β (TGF-β) in podocytes, contributing to fibrosis [6].

The largest amount of phosphorus is reabsorbed in the proximal renal tubule through the synergic action of multiple hormones, such as parathyroid hormone (PTH), fibroblast growth factor 23 (FGF23) and IGF-1 [3]. This aspect is highly important during childhood when growth occurs and IGF-1 actions on sodium-phosphate transporters (Na-Pi2a and 2c) play a crucial role [7]. Calcium and vitamin D concentrations are essential for bone growth, and the kidney tubule is one of the main sites where their concentration is regulated [8]. GH and IGF-1 increase renal calcium reabsorption by stimulating the action of 1α-hydroxylase, thus contributing to the mineralization of bone [3].

Acromegaly can be associated with nephromegaly [9]. Experiments conducted on mice demonstrated that high circulating levels of GH led to 3.5-fold renal hypertrophy, mainly due to important glomerulomegaly and interstitial edema, but with no increase in tubule diameter [10]. In addition, exposure to high levels of GH and IGF-1 for an extended period of time can lead to multiple comorbidities, such as diabetes mellitus (DM) and hypertension, which have a negative impact on kidney function [11]. Both DM and acromegaly can be associated with albuminuria, which is commonly used as a biomarker for diabetic nephropathy (DN) [9,12]. DN is caused by complex mechanisms that involve glomerular hyperfiltration, high levels of cytokines, modifications of renal gene transcription and oxidative stress, due to hyperglycemia [13]. It is characterized by glomerular basement membrane (GBM) thickening and glomerular sclerosis [13]. It was suggested that GH might be involved in the onset of DN, which is the main cause of end-stage kidney disease in developed countries [6,14].

Several data underline that novel podocyte and tubular biomarkers might precede microalbuminuria, being useful in detecting early stages of DN [15,16].

The current study aimed to investigate the effects of acromegaly on kidney function and morphology by using classic and novel renal biomarkers in order to reveal a possible correlation between GH excess and renal pathology.

## 2. Materials and Methods

### 2.1. Patients and Study Design

This case–control study was approved by the Ethics Committee of the County Emergency Hospital Timisoara (approval number 253/15.06.2021) and was conducted in accordance with the Declaration of Helsinki.

The study group was represented by the acromegalic patients admitted to the Endocrinology Department between October 2021 and November 2022. The control group included healthy subjects (without acromegaly or renal disease), matched for age, gender and body mass index (BMI).

*Inclusion criteria for the patients from the study group* were age at diagnosis >18 years, nadir GH >0.4 ng/mL during oral glucose tolerance test (OGTT) in non-diabetic patients and the increased level of IGF-1 for age and gender. In patients with DM, a mean value of 4 determinations of GH over 24 h was used instead of the OGTT, and a mean value >1 ng/mL confirmed the diagnosis of acromegaly.

*Exclusion criteria for the patients with acromegaly* were the absence of medical records confirming the diagnosis and lack of medical follow-up after the diagnosis.

*Disease status* of acromegaly was evaluated according to the Romanian guidelines [17]. Controlled disease was defined by the following: controlled symptoms; GH level <1 ng/mL, determined from a random sample or GH during OGTT <0.4 ng/mL; and IGF-1 level 1–1.3 × upper limit of normal (ULN) for age and gender. Partial disease control was defined as follows: controlled symptoms; GH level >1 ng/mL, determined from a random sample but reduced by ≥50% in comparison to its baseline value or by the IGF-1 value >1.3 × ULN but reduced by ≥50% in comparison to its value before therapy. The disease was uncontrolled if one of the following criteria were fulfilled: symptoms specific for acromegaly, basal serum GH >1 ng/mL (random sample) that did not decrease by ≥50% compared to the basal value or IGF-1 >1.3 × ULN that did not decrease by ≥50% from baseline or tumor progression. Normal values for GH and IGF-1, according to age and gender, were provided by the laboratory.

*The disease duration* was calculated from the occurrence of the first symptom or the onset of the first known complication of acromegaly. This information was collected from the patient’s medical records or personal history, if available.

All patients were screened for comorbidities associated with acromegaly. In consequence, electrocardiogram, cardiac ultrasonography, polysomnography and abdominal ultrasound were performed. For the abdominal ultrasound, a commercially available real-time machine was used. Male patients were submitted to a prostatic exam. Patients who were overweight or obese received standard advice regarding diet and lifestyle changes. They did not receive pharmacotherapy for obesity.

For the group of patients with acromegaly, the following *samples* were collected: blood (10 mL) for measuring serum creatinine, urea, blood glucose and glycated hemoglobin; and urine samples for the measurement of proteinuria/24 h, urinary albumin, urinary creatinine, urine cultures, urine electrolytes, glycosuria, nephrin and kidney injury molecule 1 (KIM-1). A reliable standardized assay determined GH and IGF-1 through chemiluminescence in the age- and gender-adjusted range.

*The inclusion criteria for the control group* were age between 18 and 70 years and the absence of known serious diseases.

*The exclusion criteria for the control group* were chronic diseases such as hypertension, DM or end-stage diseases (cardiac, neurologic, pulmonary, gastrointestinal, hematologic or malignant diseases); any acute or psychiatric disease that impairs judgment; known kidney diseases; and pregnancy.

For the control group, the following *samples* were obtained: blood (10 mL) for measuring serum creatinine and blood glucose; and urine samples to determine urinary albumin, urinary creatinine, nephrin and KIM-1. An abdominal ultrasound was performed to exclude renal abnormalities.

*The data collected for both groups* also included personal history, family history, physical examination and medical therapy.

Nephrin and KIM-1 were determined in specimens frozen at −80 °C and thawed before assay. Chronic kidney disease (CKD) was defined according to the Kidney Disease Improving Global Outcomes (KDIGO) 2024 Guideline for the Evaluation and Management of CKD [18]. The eGFR was estimated using the CKD-EPI 2021 update [19].

KIM-1 was assessed in the second-morning urine specimen by the KIM-1 ELISA test kit to detect KIM-1 in human urine, Cat No. E-EL-H6029, Elabscience (Elabscience Biotech Co., Ltd., Wuhan, Hubei Province, China). A human KIM-1 antibody was utilized, and the detection range was 7.81–500 pg/mL. The sensitivity of the assessment showed that the minimum detectable dose of human KIM-1 is 4.69 pg/mL. The repeatability of the test displayed a coefficient of variation (CV) <10%. Nephrin was assessed in the second-morning urine specimen by a human NPHN (Nephrin) ELISA kit, Cat No. E-EL-H1901, Elabscience (Elabscience Biotech Co., Ltd., Wuhan, Hubei Province, China). The sensitivity of the assessment showed that the minimum detectable dose of human NPHN is 0.1 ng/mL. The detection range is 0.16–10 ng/mL. The repeatability of the test displayed a CV <10%.

Albuminuria was measured in the second-morning urine specimen through nephelometry, using the Atellica NEPH 630 system (Siemens Healthcare Diagnostics, Marburg, Germany). Microalbuminuria was defined by a urine albumin-to-creatinine ratio (UACR) between 30 and 300 mg/g, and normoalbuminuria by a UACR <30 mg/g.

### 2.2. Statistical Analysis

Continuous variables are presented as mean value ± standard deviation (SD) or median (range). The statistical analysis was performed using MedCalc version 22.007. The following statistical tests were used: unpaired *t*-test or Mann–Whitney (for variables without Gaussian distribution) for the significance of differences, Spearman correlation to measure the strength and direction of association between two ranked variables, and ANOVA or Kruskal–Wallis to determine whether there were any statistically significant differences concerning renal parameters between patients with controlled disease versus those in which control of the disease was not achieved. The significance level was set at *p* < 0.05.

## 3. Results

The study group included 23 acromegalic patients (9 men and 14 women), aged between 19 and 72 years, with mean disease duration of 10.7 ± 7.1 years (0.1–25 years). Seventeen patients (73.9%) were under medical treatment with cabergoline, somatostatin analogs, GH receptor antagonists or a combination of these. Surgery cured acromegaly in three patients, while two patients were recently diagnosed and did not receive treatment, and one patient had a recurrence of the disease. The mean duration of treatment was 7.3 ± 5 years (0.5–16 years). The control group included 21 patients (7 men and 14 women) aged between 27 and 63 years.

From the acromegalic group, only one patient was diagnosed with CKD stage G3bA1 KDIGO, and this was considered a complication of chronic pyelonephritis and type 2 DM. Abdominal ultrasound did not reveal nephromegaly in any of the acromegalic patients. In the acromegalic group, the creatinine levels were higher and eGFR was lower than those in the control group (Table 1). It was noticed that urinary albumin and UACR were higher in patients diagnosed with acromegaly versus the patients in the control group (Table 1). There were no differences regarding renal parameters between men and women.

Acromegaly patients did not have a higher risk of increased urinary albumin excretion (Table 2). No significant differences were identified between study group and control subjects regarding urinary KIM-1/creat or urinary nephrin/creat (Table 2). Normal values for urinary KIM-1/creat and urinary nephrin/creat were estimated according to data published in the literature [20,21].

A high proportion of the acromegalic patients were diagnosed with abnormalities in glucose metabolism (78.25%) and with arterial hypertension (65.21%). There was no significant correlation between clinical or biochemical parameters associated with acromegaly and nephrin (Table 3) or KIM-1 (Table 4).

Regarding disease control, 52.17% of the patients achieved control, 21.73% had partially controlled disease, and 26.08% were uncontrolled at the time of the study. Disease control did not influence renal biomarkers, the urinary excretion of electrolytes or urinary protein excretion (Table 5).

## 4. Discussion

The primary findings of the current study point out that acromegaly is associated with higher levels of UACR and with lower levels of eGFR when compared to healthy subjects. The GH receptor is expressed in the glomerulus (mesangial cells and podocytes) and proximal tubule [2]. Podocytes and mesangial cells are important components of the nephron, and the injury of the mesangium is a common denominator of many renal pathologies [22]. GH induces collagen production, which might contribute to renal sclerosis and the alteration of the GBM [6]. Another mechanism that might contribute to renal damage is glomerular hyperfiltration, which might also be induced by high levels of IGF-1, through the increase in nitric oxide production [23]. The exposure of human podocytes to high levels of GH was associated with increased podocyte permeability to albumin [2]. There are a few cases cited in the literature that described a temporal relationship between the severity of segmental glomerulosclerosis and acromegaly [24,25]. In other studies, the renal disease ameliorated after the removal of the pituitary tumor, and it was suggested that the improvement of the hormonal profile might have contributed to the partial remission of kidney disease [25,26]. A study published by Baldelli et al. revealed that active acromegaly is associated with microalbuminuria, which was correlated with insulin resistance [27]. Untreated acromegaly was also associated with high levels of N-acetyl-beta-glucosaminidase (NAG) and glycosaminoglycans (GAGs) even though the renal function was not modified [28]. We hypothesized that prolonged exposure to supraphysiological levels of GH might weaken the GBM and lead to glomerular sclerosis, decreased eGFR, as well as albuminuria. In our study, 73% of the patients achieved disease control which might have contributed to normoalbuminuria and normal creatinine levels, similar to the findings of Auriemma et al. [9]. On the other hand, the higher levels of these markers, compared to the healthy subjects, might reflect a degree of renal impairment as a consequence of dysglycemia.

Kidney complications in acromegaly may develop due to several factors [23]. DM is diagnosed in 30% of patients [29]. Insulin resistance is the core element that contributes to abnormal glucose metabolism [30]. Several risk factors, such as older age, higher BMI, and higher levels of GH and IGF-1 were identified to favor the development of secondary DM [31,32], the latter being known as a leading cause of CKD [33].

One of the earliest consequences of DM is DN, which causes significant changes in the structure of the glomerular filtration barrier [13]. A commonly used marker for the detection of kidney damage is UACR [34]. UACR was 7.5 mg/g in our healthy subjects, while in the acromegalic group, it was 14.8 mg/g. According to the KDIGO criteria, levels below 30 mg/g are defined as being in the range of normoalbuminuria [18]. However, some studies point out that even “high-to-normal” levels of albuminuria (10–30 mg/g) are associated with increased renal and cardiovascular risk [35,36,37]. This could mean that patients with acromegaly are at higher risk of developing kidney and/or cardiac diseases.

A recent study conducted by Hong et al. concluded that acromegaly was associated with an increased risk of developing end-stage kidney disease as a result of intricated mechanisms, which include hypertension and DM [38].

A consequence of structural damage to the kidney is the detection of an increased number of podocytes and their proteins in the urine of patients with DM [39]. These proteins can be used as biomarkers in the detection of early diabetic kidney disease [40,41]. One of the most frequently used markers is nephrin [41]. It is a transmembrane glycoprotein that contributes to the physical barrier of the nephron, which prevents protein loss [42]. It seems to be a better marker than UACR due to its association with the structural damage of podocytes, as opposed to albuminuria, which is present when all three components of the glomerular filtration barrier are injured [41]. In our study, nephrin was not increased in patients with acromegaly as compared to the healthy control group. This might be explained by the fact that patients with acromegaly may not have displayed kidney damage at the moment of the study, a fact translated by normoalbuminuria. Another factor that could have contributed was the small number (only five) of patients with DM. There are studies that pointed out that high levels of nephrin in normoalbuminuric diabetic patients are not always present and, in some of them, nephrin correlated with the degree of albuminuria [20]. Current data suggest that high levels of GH lead to glomerulosclerosis [43]. Nevertheless, the interactions between GH and IGF-1 in the kidney are intricate and can be divergent [10]. In a study where podocyte dysfunction was induced through hyperhomocysteinemia, GH limited glomerulosclerosis by inhibiting the nicotinamide adenine dinucleotide phosphate (NADP) pathway [44]. IGF-1 plays a similar, protective role by inhibiting the apoptosis of podocytes [45]. On the other hand, it has been shown that exposure of the actin cytoskeleton of the podocytes to very high levels of GH leads to its reorganization [46]. The increased production of oxygen-reactive species was also observed, as a consequence of the exposure of the podocytes to high concentrations of GH [46]. It was suggested that oxidative stress might be a contributing factor to renal dysfunction in DM [47]. Due to the complex nature of these interactions, it is difficult to precisely estimate the impact of acromegaly on the glomerulus. It should be mentioned that such very high levels of GH are not common in acromegaly and the studies were conducted on cell cultures and animal models. Moreover, it is often difficult to establish the precise time when acromegaly started, so it is difficult to assess the levels of GH and IGF-1 to which the patient was exposed during the evolution of the disease.

KIM-1 is a transmembrane glycoprotein, mainly present in the tubular epithelial cells of the kidney [48]. It participates in regulating immune cells and can preserve albumin, which is reabsorbed in cases of severe proteinuria [49]. Increased levels of this protein in the urine are associated with injury of the renal proximal tubules and were described in conditions such as focal glomerulosclerosis and diabetic or hypertensive nephropathy [50]. In patients diagnosed with type 2 DM, high levels of urinary KIM-1 are predictive of a faster decline in eGFR [51]. In our study, there were no statistically significant differences in KIM-1 values between healthy volunteers and acromegalic patients.

These results might be a consequence of disease remission and the fact that GH does not cause changes in the cellular morphology of the renal tubules, but rather controls the absorption of water, sodium, and phosphate [3]. Also, Gohda et al. found that urinary KIM-1 and eGFR are better correlated in patients with chronic kidney disease (eGFR ≤ 30 mL/min/1.73 m^2^) and might reflect the recent deterioration of proximal tubular cells [40].

The actions of the GH/IGF-1 axis on the tubules are related to electrolyte homeostasis and gluconeogenesis [21]. Hypertension is diagnosed frequently (18–60%) in acromegalic patients and is characterized by elevated diastolic blood pressure [52]. The mechanisms involved are high levels of sodium reabsorption in the kidney, along with water retention [52]. Comorbidities, such as cardiac hypertrophy and sleep apnea, can contribute to increases in blood pressure, which can further deteriorate kidney function [53,54]. In our study, more than 60% of the patients were diagnosed with hypertension in different stages, but blood pressure was controlled with medication. Angiotensin-converting enzyme inhibitors and angiotensin receptor blockers were the first choice in all the patients. Both drugs favor decreases in proteinuria and might have influenced the levels of UACR, as described by other authors, as well [55]. There were no correlations between nephrin, KIM-1, UACR, eGFR, serum creatinine, GH, IGF-1, disease duration, blood pressure or hemoglobin A_1c_. We consider these results to be influenced by the significant number of acromegalic patients with controlled disease. Somatostatin analogs were the first treatment choice for our patients. It is suggested that they decrease renal blood flow and, consequently, eGFR [56]. However, in a recent study, therapy with somatostatin analogs did not influence progression to end-stage kidney disease [38]. In summary, the effects of this class of drugs on kidney function is still a matter of debate.

Even though the values of serum creatinine and UACR were higher and those of eGFR were lower in patients with acromegaly compared to healthy subjects, they were not suggestive of clinically significant kidney damage. This is also supported by the normal values of KIM-1 and nephrin. Furthermore, there was no association between acromegaly and renal biomarkers such as microalbuminuria, urinary KIM-1/creatinine or urinary nephrin/creatinine.

There are some limitations of this study. First, the small number of patients might lower the power of the statistical tests and lead to an increased variability in KIM-1 values. Second, the number of patients with uncontrolled/partially controlled acromegaly, in whom one may expect renal changes, as demonstrated by previous studies, was reduced, as well. Third, angiotensin-converting-enzyme inhibitors and angiotensin receptor blockers might have introduced a bias in the interpretation of data. However, our study has its strengths. This is one of the few studies that has assessed specific renal biomarkers of glomerular injury and proximal tubule dysfunction in acromegalic patients. Although acromegaly is historically considered a condition that affects the kidneys [9,28], and among the comorbidities, we often find hypertension and glucose metabolism disorders [11], which worsen kidney damage, in our cohort of patients, even in those with uncontrolled disease, we did not detect significant renal impairment. The explanation for this might be the very good metabolic and blood pressure control or a possible beneficial effect of the medication used to treat acromegaly.

## 5. Conclusions

Acromegalic patients presented higher levels of UACR and lower levels of eGFR, as compared with healthy subjects, even though levels were in the normal range according to current guidelines. There was no glomerular or proximal tubular damage at the time of the study, as proven by the normal levels of the kidney biomarkers nephrin and KIM-1. However, due to the complex relationship between acromegaly and kidney functionality, studies including more patients with uncontrolled disease are needed to clarify this issue.

## Figures and Tables

**Table 1 diseases-12-00211-t001:** Characteristics of the patients from the study and control groups.

Characteristic	Study Group	Control Group	*p*
Number of patients	23	21	-
Men/women	9/14	7/14	0.760
Age (years)	50.6 ± 12.4	47 ± 0.5	0.307
BMI (kg/m^2^)	32.0 ± 7.0	28.4 ± 5.7	0.069
Serum creatinine (mg/dL)	0.9 ± 0.2	0.7 ± 0.1	0.012
eGFR (mL/min/1.73 m^2^)	90.0 ± 22.2	105.5 ± 9.6	0.005
Urinary albumin (mg/L)	16.6 ± 14.0	10.4 ± 5.2	0.028
UACR (mg/g)	14.8 ± 11.8	7.5 ± 3.1	0.016
Nephrin (ng/mL)	13.0 ± 1.8	13.0 ± 1.1	0.769
Urinary nephrin/creat (mg/g)	0.012 ± 0.006	0.011 ± 0.007	0.449
KIM-1 (pg/mL)	370.6 ± 233.9	229.1 ± 265.6	0.082
Urinary KIM-1/creat (ng/g)	323.3 ± 340.7	265.6 ± 318.8	0.566

Legend: BMI = body mass index; eGFR = estimated glomerular filtration rate; UACR = urinary albumin-to-creatinine ratio; KIM-1 = kidney injury molecule 1; all data are presented as mean ± SD; create = creatinine.

**Table 2 diseases-12-00211-t002:** Association between renal biomarkers and acromegaly.

	OR	95% CI	*p*
UACR-A2	0.200	0.009–4.419	0.489
Urinary KIM-1/creat (ng/g)	0.885	0.225–3.483	1.000
Urinary nephrin/creat (mg/g)	0.348	0.013–9.047	1.000

Legend: OR = odds ratio; CI = confidence interval; creat =creatinine; UACR-A2 = urinary albumin-to-creatinine ratio—category A2 (microalbuminuria); KIM-1 = kidney injury molecule 1.

**Table 3 diseases-12-00211-t003:** Correlation between clinical and biochemical parameters and nephrin.

Parameter	r Value	*p*
Age (years)	−0.383	0.070
Estimated diagnostic delay (years)	0.039	0.857
IGF-1 at diagnosis (ng/mL)	−0.028	0.896
GH at diagnosis (ng/mL)	0.095	0.663
IGF-1 (ng/mL)	0.359	0.091
n GH (ng/mL)	0.154	0.482
IGF-1 ULN	0.343	0.108
Disease duration (years)	−0.014	0.948
Serum calcium (mg/dL)	−0.115	0.600
Urinary calcium (mg/24 h)	0.134	0.550
25(OH)vitamin D (ng/mL)	0.141	0.529
Systolic blood pressure (mmHg)	0.049	0.824
Diastolic blood pressure (mmHg)	0.021	0.922
HbA_1c_ (%)	−0.141	0.518
UACR (mg/g)	−0.080	0.714
eGFR (mL/min/1.73 m^2^)	0.211	0.333

Legend: IGF-1 = insulin growth factor-1; GH = growth hormone; n GH = nadir growth hormone; ULN = upper limit of normal; UACR = urinary albumin-to-creatinine ratio; eGFR = estimated glomerular filtration rate; HbA_1c_ = hemoglobin A_1c_.

**Table 4 diseases-12-00211-t004:** Correlation between clinical and biochemical parameters and KIM-1.

Parameter	r Value	*p*
Age (years)	−0.325	0.129
Estimated diagnostic delay (years)	0.043	0.842
IGF-1 at diagnosis (ng/mL)	−0.060	0.784
GH at diagnosis (ng/mL)	0.283	0.187
IGF-1 (ng/mL)	0.284	0.188
n GH (ng/mL)	0.221	0.309
IGF-1 ULN	0.300	0.164
Disease duration (years)	−0.253	0.242
Serum calcium (mg/dL)	0.171	0.434
Urinary calcium (mg/24 h)	0.049	0.826
25(OH)vitamin D (ng/mL)	0.354	0.106
Systolic blood pressure (mmHg)	0.208	0.339
Diastolic blood pressure (mmHg)	0.188	0.390
HbA_1c_ (%)	0.180	0.411
UACR (mg/g)	−0.304	0.158
eGFR (mL/min/1.73 m^2^)	0.278	0.198

Legend: KIM-1 = kidney injury molecule 1; IGF-1 = insulin growth factor-1; GH = growth hormone; n GH = nadir growth hormone; UACR = urinary albumin-to-creatinine ratio; eGFR = estimated glomerular filtration rate; ULN = upper limit of normal; HbA_1c_ = hemoglobin A_1c_.

**Table 5 diseases-12-00211-t005:** Correlations between disease control and renal biomarkers, urinary electrolytes, proteinuria and eGFR.

Parameter	Disease Control	Number of Patients	Mean/SD	*p*
KIM-1, (pg/mL)	Uncontrolled	6	453.9 ± 154.2	0.592
	Controlled	12	330.0 ± 239.0	
	Partial control	5	368.2 ± 314.2	
Urinary KIM-1/creat (ng/g)	Uncontrolled	6	316.8 ± 102.5	0.801
	Controlled	12	340.8 ± 290.7	
	Partial control	5	442.8 ± 554.6	
Nephrin, (ng/mL)	Uncontrolled	6	13.3 ± 0.7	0.223
	Controlled	12	12.4 ± 2.2	
	Partial control	5	14.0 ± 0.7	
Urinary nephrin/creat (mg/g)	Uncontrolled	6	0.009 ± 0.001	0.346
	Controlled	12	0.013 ± 0.006	
	Partial control	5	0.015 ± 0.009	
Serum creatinine, (mg/dL)	Uncontrolled	6	0.7 ± 0.2	0.247
	Controlled	12	0.9 ± 0.2	
	Partial control	5	0.8 ± 0.8	
eGFR (mL/min/1.73 m^2^)	Uncontrolled	6	91.2 ± 15.8	0.701
	Controlled	12	79.9 ± 22.2	
	Partial control	5	105.8 ± 18.9	
Urinary albumin, (mg/L)	Uncontrolled	6	26.2 ± 24.8	0.204
	Controlled	12	12.8 ± 4.7	
	Partial control	5	14.4 ± 9.0	
Urinary creatinine, (mg/dL)	Uncontrolled	6	144.6 ± 29.3	0.498
	Controlled	12	114.3 ± 56.0	
	Partial control	5	182.0 ± 219.1	
UACR, (mg/g)	Uncontrolled	6	19.7 ± 20.3	0.868
	Controlled	12	13.5 ± 7.3	
	Partial control	5	12.0 ± 7.6	
Urinary calcium, (mg/24 h)	Uncontrolled	6	128.3 ± 80.1	0.733
	Controlled	12	139.7 ± 100.4	
	Partial control	5	172.4 ± 99.2	
Urinary phosphorus, (g/24 h)	Uncontrolled	6	0.6 ± 0.1	0.743
	Controlled	12	0.7 ± 0.3	
	Partial control	5	0.7 ± 0.2	
Urinary sodium, (mmol/24 h)	Uncontrolled	6	110.8 ± 53.3	0.41
	Controlled	12	107.1 ± 51.0	
	Partial control	5	142.0 ± 35.8	
Urinary proteins, (g/24 h)	Uncontrolled	6	0.1 ± 0.2	0.619
	Controlled	12	0.1 ± 0.1	
	Partial control	5	0.1 ± 0.0	

Legend: SD = standard deviation; KIM-1 = kidney injury molecule 1; UACR = urinary albumin-to-creatinine ratio; eGFR = estimated glomerular filtration rate; creat = creatinine.

## Data Availability

The data presented in this study are available on request from the corresponding author.

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
