# Peer review of "The Use of Kidney Biomarkers, Nephrin and KIM-1, for the Detection of Early Glomerular and Tubular Damage in Patients with Acromegaly: A Case–Control Pilot Study"

_diseases, 2024, doi:10.3390/diseases12090211_

Round 1

Reviewer 1 Report

Comments and Suggestions for Authors

The manuscript presents the results of investigation of the effects of acromegaly on kidney function and morphology,. Authors used classic and novel renal biomarkers KIM-1 and nephritis.

The title and the aim are interesting however, the description of the materials and method needs more details:

1.How long the acromegaly was existed and how long patients were treated?

2.Did any differences occurred between F and M?

3.Did the patients were treated against the obesity (high BMI)?

4. What was the reason of high GH level?

Description of acromegaly needs more details, stating that the main reason of this disorder is high level of growth hormone is not enough (Introduction).

Without  answers to above questions the conclusions are not proper.

In summary: it is suggestion to add these information.

Author Response

  1. How long the acromegaly was existed and how long patients were treated?

Response: Mean disease duration in our group was 10.7 ± 7.1 years (see Results – row 172). We added the type of treatment (see Results – rows 172-176). The duration of therapy is different for each patient. Mean duration of treatment was added in the paper (see Result – rows 176-177).

  1. Did any differences occurred between F and M?

Response: No differences were noted between males and females. (This information was added at rows 185-186).

  1. Did the patients were treated against the obesity (high BMI)?

Response: They received standard advice regarding diet and lifestyle changes. They did not receive farmacotherapy for obesity (This information was added in the paper at rows 124-125).

  1. What was the reason of high GH level? Description of acromegaly needs more details, stating that the main reason of this disorder is high level of growth hormone is not enough (Introduction).

Response: We added information in Introduction – rows 45-47.

Reviewer 2 Report

Comments and Suggestions for Authors

The authors present a manuscript proposing nephrin and KIM-1 as biomarkers related with nephropathy and acromegaly. The term “involvement” in the title of the article can be modified to reflect the parameters of kidney damage that were found in acromegaly but clarifying that nor nephrin nor KIM-1 were correlated. The acromegaly group includes patients with “controlled disease”, but there is no information about the therapeutics received, which might be involved in the prevention of kidney damage. The authors do comment about the use of angiotensin-converting-enzyme inhibitors and angiotensin receptor blockers that decrease proteinuria. This reviewer suggest that the manuscript can be focused on the no-damaged kidney in controlled acromegaly, confirmed with the lack of changes for the biomarkers that were tested; the authors have considered in the conclusions the limitations of the study, but did not discuss that activation of GH receptor increases the glomerular filtration rate (it is increased in patients with active acromegaly). Finally, the authors might revise their ELISAs, considering that standard deviation is greater than the mean itself.

Comments on the Quality of English Language

There are minor details to be considered: physiologic (physiological), basement, high levels of podocytes

Author Response

The authors present a manuscript proposing nephrin and KIM-1 as biomarkers related with nephropathy and acromegaly.

The term “involvement” in the title of the article can be modified to reflect the parameters of kidney damage that were found in acromegaly but clarifying that nor nephrin nor KIM-1 were correlated.

Response: We have modified the title according to your suggestion.

The acromegaly group includes patients with “controlled disease”, but there is no information about the therapeutics received, which might be involved in the prevention of kidney damage.

Response: The majority of the patients were under medical treatment with somatostatin analogs. There are conflicting results in the literature regarding somatostatin analogs and their influence on kidney function. (We added information in Discussion – rows 319-323).

The authors do comment about the use of angiotensin-converting-enzyme inhibitors and angiotensin receptor blockers that decrease proteinuria. This reviewer suggest that the manuscript can be focused on the no-damaged kidney in controlled acromegaly, confirmed with the lack of changes for the biomarkers that were tested; the authors have considered in the conclusions the limitations of the study, but did not discuss that activation of GH receptor increases the glomerular filtration rate (it is increased in patients with active acromegaly).

Response: In our introduction, we have mentioned the following: “The GH receptor is expressed in the podocytes, mesangial cells, and tubules, where it maintains normal sodium and water balance (probably through its action on the epithelial sodium channel) and increases the glomerular filtration rate (eGFR)” (rows 56-59).

We also added in Discussion a comment regarding this issue.

Finally, the authors might revise their ELISAs, considering that standard deviation is greater than the mean itself.

Response: Indeed, some results showed a high standard deviation, due to the variability of the individual results and the small study group. This was recognized as a limitation of our study.

There are minor details to be considered: physiologic (physiological), basement, high levels of podocytes

Response: We corrected these mistakes (highlighted in the text), as suggested.

Reviewer 3 Report

Comments and Suggestions for Authors

Dear editors:  

 It is a great honor and pleasure for me to be invited as the reviewer for this research entitled “Kidney biomarkers nephrin and KIM-1 for the detection of early kidney involvement in patients with acromegaly: a case-control pilot study”. Iulia Stefania Plotuna and coauthors investigated the clinical value of urinary biomarkers nephrin and KIM-1 in the early detection of AKI for acromegalic patients. This study topic is interesting and important, attributing to their team’s long-term efforts in the scientific field. Although the manuscript was well-written, I have a small number of comments:

1.     The scope of the study focused on the clinical value of nephrin and KIM-1 to detect “early kidney involvement” in patients with acromegaly. However, biased and unrepresentative sample was noted in Table 1:

Serum creatinine (mg/dL) 0.9 ± 0.2 vs. 0.7 ± 0.1; eGFR (mL/min/1.73m2) 90.0 ± 22.2 vs.105.5 ± 9.6; UACR (mg/g) 14.8 ± 11.8 vs.7.5 ± 3.1.

In fact, all patients exert normal renal function in both Study group and Control group. In light of KDIGO guideline, the kidney injury could not be found in this G1A1 stage. Thus the results were insignificant.

2.     To explore the aim of the study, authors should select acromegalic patients with albumiuria for second analysis of nephrin and KIM-1: acromegalic patients with albumiuria, acromegalic patients without albumiuria, control group.

3.     To expamd the aim of the study, authors should follow those acromegalic patients with higher levels of UACR, nephrin and KIM-1.

Author Response

Dear editors:

It is a great honor and pleasure for me to be invited as the reviewer for this research entitled “Kidney biomarkers nephrin and KIM-1 for the detection of early kidney involvement in patients with acromegaly: a case-control pilot study”. Iulia Stefania Plotuna and coauthors investigated the clinical value of urinary biomarkers nephrin and KIM-1 in the early detection of AKI for acromegalic patients. This study topic is interesting and important, attributing to their team’s long-term efforts in the scientific field. Although the manuscript was well-written, I have a small number of comments:

  1. The scope of the study focused on the clinical value of nephrin and KIM-1 to detect “early kidney involvement” in patients with acromegaly. However, biased and unrepresentative sample was noted in Table 1:

Serum creatinine (mg/dL) 0.9 ± 0.2 vs. 0.7 ± 0.1; eGFR (mL/min/1.73m2) 90.0 ± 22.2 vs.105.5 ± 9.6; UACR (mg/g) 14.8 ± 11.8 vs.7.5 ± 3.1.

In fact, all patients exert normal renal function in both Study group and Control group. In light of KDIGO guideline, the kidney injury could not be found in this G1A1 stage. Thus the results were insignificant.

Response: Indeed, this is an important issue and we have addressed it: “Even though the values of serum creatinine and UACR were higher and those of eGFR lower, in patients with acromegaly compared to healthy subjects, they were not suggestive of clinically significant kidney damage. This is also supported by the normal values of KIM-1 and nephrin“ (see Discussion – rows 324-327).

  1. To explore the aim of the study, authors should select acromegalic patients with albumiuria for second analysis of nephrin and KIM-1: acromegalic patients with albumiuria, acromegalic patients without albumiuria, control group.

Response: In our current cohort of patients, only two met the criteria for increased urinary albumin excretion. Due to this small number, statistical tests could not be performed. However, we consider your suggestion and plan to expand the study, including a higher number of patients with increased urinary albumin excretion.

  1. To expamd the aim of the study, authors should follow those acromegalic patients with higher levels of UACR, nephrin and KIM-1.

Response: Thank you for your comment. We plan to expand the study, having the following objectives:

- to increase the cohort of patients with active acromegaly and assess renal function in them;

- to find factors that might predict the risk of renal impairment in acromegaly;

- to use other kidney makers, such as cystatin C, to better evaluate renal function.

Round 2

Reviewer 1 Report

Comments and Suggestions for Authors

I do not have further comments

Author Response

Thank you for the time spent to review our paper.

Reviewer 2 Report

Comments and Suggestions for Authors

The authors have revised the manuscript; they have included changes for suggestions from this reviewer. Discussion in lines 319-323 could be further improved, although; the authors could analyze the results to test this statement and correlate treatments and eGFR if needed. This might validate the results for the experimental group (versus the control group) in terms of previous reports, considering eGFR. No revision or analysis was reported in the new version for the variation in KIM-1 results.

Reviewer 3 Report

Comments and Suggestions for Authors

1.Authors should revise Table 2: Comparison of renal biomarkers between acromegaly patients and control subjects.

Microalbuminuria=> UACR (mg/g)

2.Again, I strongly suggest  authors should include more acromegalic patients with albumiuria for second analysis of nephrin and KIM-1: acromegalic patients with albumiuria, acromegalic patients without albumiuria, and control group.

Renal injury markers, nephrin and KIM-1, could not be elevated in acromegalic patients without albumiuria.

Comments on the Quality of English Language

Extensive editing of English language is required.
